Genetic changes in the EPAS1 gene between Tibetan and Han ethnic groups and adaptation to the plateau hypoxic environment

Li Cuiying licuiying2017@126.com
Li Xiaowei
Xiao Jun
Liu Juan
Fan Xiu
Fan Fengyan
Lei Huifen
Department of Blood Transfusion, Air Force Medical Center, PLA , Beijing , China
Tatarinova Tatiana
Electronic publication date: 2019 Oct 28
Publication date: 2019
Volume: 7
Electronic Location ID: e7943
Received 2019 Jul 17; Accepted 2019 Sep 24
Copyright: ©2019 Li et al.
Copyright year: 2019
Copyright holder: Li et al.
License: This is an open access article distributed under the terms of the Creative Commons Attribution License, which permits unrestricted use, distribution, reproduction and adaptation in any medium and for any purpose provided that it is properly attributed. For attribution, the original author(s), title, publication source (PeerJ) and either DOI or URL of the article must be cited.
License URL: https://creativecommons.org/licenses/by/4.0/

Keywords: Plateau, Hemoglobin, EPAS1, Raman spectroscopy

Funding: Military Major Special Project of PLA, China AWS13J004 Key Logistics Research Projects of PLA, China BWS16J006-03-01 This work was supported by the Military Major Special Project of PLA, China (No. AWS13J004) and the Key Logistics Research Projects of PLA, China (No. BWS16J006-03-01). The funders had no role in study design, data collection and analysis, decision to publish, or preparation of the manuscript.

==============================
In the Chinese Han population, prolonged exposure to hypoxic conditions can promote compensatory erythropoiesis which improves hypoxemia. However, Tibetans have developed unique phenotypes, such as downregulation of the hypoxia-inducible factor pathway through EPAS1 gene mutation, thus the mechanism of adaption of the Han population should be further studied. The results indicated that, under plateau hypoxic conditions, the plains population was able to acclimate rapidly to hypoxia through increasing EPAS1 mRNA expression and changing the hemoglobin conformation. Furthermore, the mutant genotype frequencies of the rs13419896, rs1868092 and rs4953354 loci in the EPAS1 gene were significantly higher in the Tibetan population than in the plains population. The EPAS1 gene expression level was lowest in the Han population carrying the A-A homozygous mutant of the rs13419896 locus but that it increased rapidly after these individuals entered the plateau. At this time, the hemoglobin content was lower in the homozygous mutant Han group than in the wild-type and heterozygous mutant populations, and the viscosity of blood was reduced in populations carrying the A-A haplotypes in rs13419896 and rs1868092 Among Tibetans, the group carrying homozygous mutations of the three SNPs also had lower hemoglobin concentrations than the wild-type. The Raman spectroscopy results showed that exposure of the Tibetan and Han population to hypoxic conditions changed the spatial conformation of hemoglobin and its binding ability to oxygen. The Tibetan population has mainly adapted to the plateau through genetic mutations, whereas some individuals adapt through changes in hemoglobin structure and function.

Introduction

The Qinghai-Tibet Plateau is located in the western part of China and has an average elevation of 4,000 m. It is characterized by low pressure, low oxygen, low temperature, high winds, dry climate, strong ultraviolet radiation and other unfavorable factors. The oxygen content is only 60% of that of the plains, and millions of annual trips are made to the Qinghai-Tibet Plateau annually, presenting a serious environmental challenge for humanity (Petousi et al., 2014). In addition, some people must live in the plateau environment for a long time. Prolonged exposure to hypoxic conditions can promote compensatory erythropoiesis, increased hemoglobin levels, and an increased blood oxygen capacity, all of which improve hypoxemia. Upon initially entering the plateau, members of the plains population may experience acute altitude sickness due to a lack of oxygen, with main manifestations of palpitation, shortness of breath, headaches, and sleep disturbances. In severe, high-altitude brain edema or pulmonary edema and death may occur (Meier et al., 2017). During a prolonged stay in the plateau, the above-mentioned symptoms of adverse reactions may ease or disappear. If the above symptoms persist for more than three months, red blood cells will continue to proliferate, and the condition can develop into chronic high plateau diseases such as high plateau polycythemia. However, Tibetan people who have lived in the plateau for generations have a unique set of plateau-adapted physiological characteristics: their arterial oxygen content is basically unchanged, resting ventilation is increased, birth weight is increased, and hemoglobin concentration is low (Bigham & Lee, 2014). Recent studies have shown that these physiological changes are closely related to the genetic characteristics of the population, highlighting the importance of genetic adaptation and not solely phenotypic changes in plateau adaptation (Bigham & Lee, 2014).

Genomics, transcriptomic microarray, and whole-genome linkage analysis have revealed that hypoxia inducible factor (HIF)-1 and HIF-2 plays an important role in altitude hypoxia adaptation (Lendahl et al., 2009). Studies have shown that HIF is widely expressed in mammals and that it plays a key role in hypoxia and can promote the adaptation of mammals to high altitude environments (Bigham & Lee, 2014). The most important study of HIF focused on endothelial PAS domain protein 1 (EPAS1), which encodes HIF-2α; several single nucleotide polymorphisms (SNPs) in EPAS1 is closely related to the low level of hemoglobin (Hb) in Tibetan populations (Beall et al., 2010; Xu et al., 2015). Specifically, rs13419896, rs1868092, and rs4953354 are great difference between Tibetan and Han populations (Peng et al., 2011), and are also significantly correlated with low hemoglobin (Beall et al., 2002; Chen et al., 2014). EPAS1 distribution is tissue-specific; it is mainly expressed in tissues and organs related to metabolism and the oxygen supply, such as placenta,vascular endothelium (Hu et al., 2003) and kidney (Lee & Percy, 2011). Therefore, it was inferred that EPAS1 expression plays an important role in oxygen metabolism and sensing.

The most striking differences between Han people who acutely enter the plateau and Tibetans whose families have lived in the plateau for generations are related to Hb. Why do some people experience sickness after acutely entering the plateau? Can the Han population show acclimation after living in the plateau for a long time? To answer these questions, we tested the physiological indicators of three groups of people. We found that there was no significant difference in Hb concentration between the Tibetans and plains population, whereas the number of red blood cells and the Hb level of people who acutely enter the plateau and those who lived in the plateau for a long time were significantly higher than those of the plains. The blood oxygen saturation level (SpO2) of people acutely entering the plateau and those who had lived in the plateau for a long time were significantly lower than that of the plains population, while the level of Tibetans was slightly lower than that of the plains population. Compared with the plains population, the affinity of Hb for oxygen (P50) increased rapidly in people who acutely entered the plateau and was lower in the Tibetan population and in those who had lived in the plateau for a long time (Li et al., 2018). To clarify the mechanism and identify detection and prevention measures for the plains population prior to entry into the plateau, we conducted a series of studies at the molecular level focusing on the genetic and epigenetic differences in the EPAS1 gene and on Hb conformations.

Materials and Methods

Materials

(1) We selected 155 Tibetans whose families had lived at high altitude (3,700 m above sea level, Lhasa City, Tibet, China) for generations. This group of individuals included 155 males aged 18–57 years, with an average age of (33.9 ± 8.3); (2) we also selected 190 Han male people whose families had lived in the plains (50–500 m above sea level) for generations; this population aged 13–55 years, (average age 34.7 ± 9.2 years); Four additional population were included in the study: (3) 50 plains Han people who remained in the plateau at 4,300 m above sea level for three days; (4) 50 plains Han people who remained in the plateau at 4,300 m above sea level for seven days; (5) 44 plains Han people who remain in the plateau at 3,700 m above sea level for 30 days; and (6) 24 Han men who had lived in the plateau for a long time (>10 years, 3,700 m above sea level). Group (3), (4), and (5) consisted of the same group of men (Table 1).

Table 1 The baseline characteristics of each group.

	No.	Age	BMI	Smoking	Altitude (m)	First exposure?	
Plains	190	24.12 ± 4.40	23.00 ± 1.12	No	500	—	
Han entered plateau 3rd day	50	23.50 ± 4.42	23.07 ± 1.16	No	4200	√	
Han entered plateau 7th day	50	19.48 ± 1.25	22.29 ± 1.35	No	4,200	√	
Han entered plateau 30th day	44	22.70 ± 3.60	22.25 ± 1.50	No	3,700	√	
The plateau Han population (living in plateau>10years)	24	41.50 ± 7.60	22.63 ± 1.39	No	3,700	—	
Tibetan	155	35.31 ± 15.53	23.16 ± 1.76	No	3,700	√	

This study was approved by the Institutional Review Board of Air Force Medical Center, PLA (2017-05-YJ01) and the research was carried out in accordance with the World Medical Association Declaration of Helsinki. All volunteers signed informed consent forms, and personal background investigations were conducted to rule out kinship among them.

Monitoring of routine physiological indicators

Routine blood examination was performed for 155 subjects in group (1), 190 subjects in group (2), 50 subjects in group (3), 50 subjects in group (4), 44 subjects in group (5), and 24 subjects in group (6) using the automatic whole blood cell analyzer (Model TEK3600, Tecom Science Co., Ltd., China). For 50 of the subjects in group (1) through (3), 44 of the subjects in group (5) and 10 of the subjects in group (6), the blood oxygen saturation level was measured with a portable blood oxygen saturation detector.

Detection of mutation loci in the EPAS1 gene

DNA was isolated from 200 µL of EDTA-anticoagulated peripheral blood collected from each of 100 subjects in group (1), 100 subjects in group (2), and 50 subjects in group (6). Genomic DNA was extracted using a blood genomic DNA extraction kit according to directions provided with the kit (Tiangen Bio Inc., Beijing, China).

Design of the primer sequence corresponding to the EPAS1 gene sequence was based on information obtained from NBCI. The following primers were designed and used in amplification and sequencing based on the information for the rs13419896, rs1868092, and rs4953354: F1: 55′-TCATTCCCTGTTCCCTCCTCCTT-3′, R1: 5′- GCCAGCTTCCCTTGACCATCTT-3′; F2: 5′-TGAGCTGATAAGACTGGTGA-3′, R2: 5′-AAGTACATGCTGCTGGAATG-3′; F3: 5′-AGAGGGAATCCAGTGTGAGG-3′, R3: 5′-GGGAGTGGTGATGAAAGAAG-3′.

The above mutation sites of the EPAS1 gene were detected by PCR and direct sequencing. The PCR reaction system consisted of 10 µL of PCR premix, 0.5 µL of each upstream and downstream primers (10 µM), 2 µL of template, and water added to a total volume of 20 µL. The reaction conditions were as follows: pre-denaturation at 95 °C for 3 min, 28 cycles of denaturation at 95 °C for 5 s, annealing at 57 °C for 20 s, and extension at 72 °C for 20 s. The resulting PCR products were sent directly to SinoGenoMax Co., Ltd, China, for sequencing.

RT-qPCR measeurement of EPAS1 gene expression level in each group

Total RNA was extracted from 400 µl of EDTA-K2 anticoagulated peripheral blood according to the procedures supplied in the manual of a Whole Blood RNA Extraction Kit (Life Technology). RNA with good purity (OD260/OD280between 1.7 and 2.0) was used in subsequent experiments.

Reverse transcription was performed using ReverTra Ace® qRCR RT Master Mix with gDNA Remover (Toyobo Co. Ltd.) in a reaction consisting of 2 µL of DNA Mix (with gRNA Remover added), 2 µL of RT Master Mix II, 500 ng of RNA template, and water added to a total volume of 10 µL. The reverse transcription reaction was performed at 37 °C for 15 min, 50 °C for 5 min, and 98 °C for 5 min, and the obtained cDNA was stored at −20 °C.

PCR primers were designed based on the information available in National Center for Biotechnology Information (NCBI) database, as follows: upstream primer: 5′-TTGATAGCAGTGGCAAGGGG-3′, downstream primer: 5′-GAGATGATGGCGTCTCCTGG-3′; upstream primer of the reference gene RPL13A: 5′-AAAAGCGGATGGTGGTTCC-3′, downstream primer of the reference gene RPL13A: 5′-GCTGTCACTGCCTGGTACTT-3′. The reaction system consisted of 10 µL of SYBR Green premix, 0.5 µL each of the upstream and downstream primers (10 µM), 2 µL of template, and water added to a total of 20 µL. The PCR program was as follows: pre-denaturation at 95 °C for 3 min and then 95 °C for 5 s, 57 °C for 20 s, and 72 °C for 20 s for 40 cycles. Melting curve analysis and PCR data analysis were performed using the Bio-Rad CFX96 program.

Raman spectroscopy analysis of Hb conformation

We selected ten individuals in group (1), three in group (2), three in group (3), three in group (4), three in group (5), and three in group (6); all of these male individuals were selected by random number generator. The red blood cells obtained from these subjects were mixed with glycerin, and the samples were loaded into a 96-well plate (approximately 200 µL/well). Raman spectrometry was set at an excitation wavelength of 514 nm, power of 2 mW, and an exposure time of 30 s and was repeated twice. The incident laser was focused on a point 300 µm below the liquid surface. The Raman spectra were measured using a 20x lens. The obtained Raman spectral data were subjected to baseline calibration using Origin 8.0 software and smoothed by the Savitzky-Golay method.

Data analysis

SPSS 20.0 statistical software was used for data processing and statistical analysis. Multiple groups were compared using one-way analysis of variance (ANOVA). Dunnett’s test was used for comparisons between groups. P < 0.05 was considered to indicated statistical significance.

Results

Differences in physiological indexes between Tibetan and Han ethnic groups in the plateau hypoxic environment

In order to reveal the mechanism of plateau adaptation difference between Tibetan and Han population, we first compared the Hb concentration changes among these two groups. Compared with the Hb level of the Han population in the plains (127–179 g/L, 152.4 ± 9.1 g/L), the Hb level of the Tibetan (120–180 g/L, 148.4 ± 15.6 g/L) did not show significant changes; however, Hb level of Han individuals who acutely entered the plateau for three days (153–197 g/L, 171.8 ± 11.0 g/L) was significantly increased. After these individuals had remained in the plateau for a prolonged period, their Hb level (150–201 g/L, 175.9 ± 10.3 g/L) remained relatively high. In Han individuals who had lived in the plateau for a long time (>10 years), the Hb concentration (124–201 g/L, 171.1 ± 9.149 g/L) was slightly decreased, but it was still higher than that of the general Han populations (Fig. 1A).

Figure 1 The differences of physiological indexes between Tibetan and Han ethnic groups in the plateau hypoxic environment.

(A) The hemoglobin level in different population. (B) The hemoglobin level in different population.*P < 0.05 or ***P < 0.001 for indicated group versus plain group. N .S: No statistical difference.

SpO2 was significantly lower in individuals who acutely entered the plateau for three days (79–93%, 87.59 ± 3.10%) than in the plains population (94–99%, 97.28 ± 1.46%). SpO2 increased slightly in individuals who had acclimated to the plateau for 30 days (85–95%, 90.36 ± 2.57%) but was still lower than that in the plains group. The SpO2 of Han individuals who had lived in the plateau for a long time (80–95%, 87.70 ± 5.56%) was similar to that of the group that acutely entered the plateau for three days. Compared with the plains group, the SpO2 of Tibetan (90–99%, 94.55 ± 2.35%) was lower than that of the plains group but significantly higher than that of other groups (not labeled) (Fig. 1b).

Differences in EPAS1 gene polymorphism, and mRNA expression between Tibetan and Han ethnic groups

As an important transcription factor of HIF-2α, EPAS1 directly activates the downstream expression of erythropoietin (EPO), which in turn leads to an increase in erythropoiesis. This is the basis for adaptation in people who initially enter the plateau (Simonson et al., 2015). EPAS1 gene polymorphism and expression are directly related to the Hb concentration and blood oxygen saturation in Tibetans who are indigenous to the plateau (Peng et al., 2017). To elucidate the mechanism and significance of the above-mentioned changes in physiological indexes under hypoxic conditions in the plateau, we further compared the genetic changes in the EPAS1 gene between the Tibetan and Han populations to elucidate the theoretical basis for the prevention and treatment of altitude sickness in the plains population.

Distribution of EPAS1 gene polymorphisms in Tibetan and Han populations

Three different SNPs of the EPAS1 gene were founded in different subpopulations. As shown in Table 2, the frequencies of the A allele of the rs13419896 locus were 86.1%, 32.6% in the Tibetans and the Han population, respectively. In these population, the frequencies of the “G-A” and “A-A” genotypes were 22.6%, 47.4% and 74.8%, 8.9%, respectively; the frequencies of the A allele of the rs1868092 locus were 72.6% and, respectively, and the frequencies of the “G-A” and “A-A” genotype were 39.4%, 19.5% and 52.9%, 0.5%, respectively; the frequencies of the G allele of the rs4953354 locus were 75.5% and 12.6%, respectively; and the frequencies of the “G-A” and “G-G” genotypes were 25.8%, 23.1% and 62.6%, 10.1%, respectively. There were statistically significant differences in the frequencies of the three SNPs between the Tibetan and Han groups. Therefore, the genotype with homozygous mutant alleles in the three SNPs is called the plateau-adaptive genotype.

Table 2 Three SNPs genotype and allele frequency distribution of EPAS1 gene in different populations.

SNP	Genotype or alleleassociated with SNP	Native Tibetans (n%. N = 155)	Plain group (n%, N = 190)	OR (95% CI)	P value*	
rs13419896						
Genotype	G/G	4 (2.6)	83 (43.7)	1		
	G/A	35 (22.6)	90 (47.4)	0.124(0.042–0.364)	0.000	
	A/A	116 (74.8)	17 (8.9)	0.007 (0.002–0.022)	0.000	
Allele	G	43 (13.9)	256 (67.4)	1		
	A	267 (86.1)	124 (32.6)	0.078 (0.053–0.115)	0.000	
rs1868092						
Genotype	G/G	12 (7.7)	152 (80.0)	1		
	G/A	61 (39.4)	37 (19.5)	0.048 (0.023–0.098)	0.000	
	A/A	82 (52.9)	1 (0.5)	0.001 (0.000–0.008)	0.000	
Allele	G	85 (27.4)	341 (89.7)	1		
	A	225 (72.6)	39 (10.3)	0.043 (0.029–0.065)	0.000	
rs4953354						
Genotype	A/A	18 (11.6)	144 (75.8)	1		
	G/A	40 (25.8)	44 (23.1)	0.138 (0.072–0.264)	0.000	
	G/G	97 (62.6)	2 (10.1)	0.003 (0.001–0.011)	0.000	
Allele	A	76(24.5)	332 (87.4)	1		
	G	234 (75.5)	48 (12.6)	0.047 (0.032–0.070)	0.000	
Notes.

OR: Odds ratio, OR >1 indicates that this factor is a risk factor; OR value <1 indicates that this factor is a protective factor.

* means Heterozygous mutation or Homozygous mutation group compared with wild type group, P < 0.05 was considered to be statistically significant.

Differences in EPAS1 mRNA expression between the Tibetan and Han populations

Real-time quantitative PCR was used to measure the EPAS1 gene expression levels in Han subjects whose families had lived in the plains for generations, Han subjects who entered and remained in the plateau for various time periods, Han subjects living in the plateau for a long time, and Tibetan. The results showed no change EPAS1 gene expression occurred in individuals who acutely entered the plateau for three days; however, the EPAS1 mRNA level was significantly increased at seven days, but decreased after 30 days of acclimation in the plateau, and increased again in individuals who had lived in the plateau for a long time (>10 years). However, there was no significant difference in EPAS1 mRNA levels between the Tibetan and Han population (Fig. 2).

Figure 2 The differences in EPAS1 mRNA expression between the Tibetan and Han populations.

mRNA was extracted from peripheral blood in different group and reversed into cDNA, and RPL13A was used to calibrate sample loading. *P < 0.05 or ***P < 0.001 for indicated group versus plain group. N.S: No statistical difference.

Effect of genetic alteration of the EPAS1 gene in the plateau hypoxic environment on the physiological indexes of Tibetan and Han peoples

Effect of EPAS1 polymorphism on Hb and blood oxygen saturation levels

Then, in order to investigate different physiological indexes in the Tibetan groups and in Han populations, stratified analysis was used based on the three genotypes in the EPAS1 gene. The results showed that Han individuals of the plains population with various genotypes showed no significant difference in Hb levels before entering the plateau. However, when the Han groups entered the plateau, the Hb concentrations of the plateau-adaptive genotype with homozygous alleles in the three SNPs were lower than those of the other two genotypes. In Han groups, the average Hb concentration of the homozygous mutant of the rs13419896 site (170 g/L) was lower than the wild-type (173 g/L) and that of the heterozygous mutant (171 g/L); the average Hb concentration of the homozygous mutant of the rs1868092 site was 164 g/L, which was significantly lower than that of the wild-type at 172 g/L and that of the heterozygous mutant at 172 g/L; the average Hb concentration of the homozygous mutant of the rs4953354 site was 169 g/L, lower than both that of the wild-type at 174 g/L and the heterozygous mutant at 174 g/L. The average Hb concentration of the plateau-adaptive genotype of the rs13419896 site in Han living in the plateau (>10 years) was 158 g/L, significantly lower than both the wild-type at 174 g/L and the heterozygous mutant at 171 g/L. In Tibetans, the Hb concentrations of the plateau-adaptive genotype with homozygous alleles in the three SNPs were lower than the wild-type and that of the heterozygous mutant; the average Hb concentration of the homozygous mutant of the rs13419896 site was 153 g/L, significantly lower than the wild-type at 172 g/L and the heterozygous mutant at 158 g/L; the average Hb concentration of the homozygous mutant at the rs1868092 site was 150 g/L, significantly lower than the wild-type at 189 g/L and the heterozygous mutant at 154 g/L; the average Hb concentration of the homozygous mutant of the rs4953354 site was 153 g/L, significantly lower than the wild-type at 160 g/L and the heterozygous mutant at 158 g/L (Table 3). These findings indicated that genetic mutations in the plains Han Chinese also have an effect on the Hb concentration. However, the association of these mutations with blood oxygen saturation did not differ among all the populations (Table 4).

Table 3 Genotype of three SNPs within the EPAS1 gene and differences in Hb levels in different populations.

	Hb (g/L) SNP sites	Wild type	Heterozygous mutation	Homozygous mutation	
Plain group	rs13419896	151.2 ± 12.2	147.6 ± 12.0	149.1 ± 9.8	
rs1868092	149.8 ± 12.3	147.5 ± 10.3	133.0 ± 0.0	
rs4953354	150.1 ± 11.2	148.0 ± 14.0	140.7 ± 12.4	
Plateau (3 days)	rs13419896	173.7 ± 12.9	171.6 ± 10.5	170.4 ± 5.3	
rs1868092	172.6 ± 11.4	172.3 ± 10.1	164.0	
rs4953354	172.4 ± 10.1	172.3 ± 14.4	172.5 ± 12.0	
Plateau (7 days)	rs13419896	174.9 ± 9.4	174.3 ± 12.7	169.7 ± 10.4	
rs1868092	170.5 ± 28.9	173.8 ± 10.7	150.0	
rs4953354	169.3 ± 30.1	175.8 ± 10.6	166.0 ± 22.6	
Plateau (30 days)	rs13419896	177.7 ± 10.9	175.7 ± 9.5	170.8 ± 11.7	
rs1868092	177.7 ± 8.7	169.4 ± 14.8	165.0	
rs4953354	176.1 ± 7.4	175.9 ± 17.0	173.0 ± 11.3	
Plateau (>10 years)	rs13419896	174.5 ± 11.3	171.1 ± 18.8	158.2 ± 19.8	
rs1868092	171.2 ± 16.3	171.3 ± 19.5	170.0 ± 6.0	
rs4953354	170.3 ± 16.1	169.4 ± 19.0	177.4 ± 11.0	
Native Tibetans	rs13419896	172.0 ± 16.8	158.9 ± 31.5	153.8 ± 27.7	
rs1868092	189.7 ± 18.0	154.7 ± 29.4	150.1 ± 25.1	
rs4953354	160.7 ± 24.3	158.3 ± 32.1	152.9 ± 28.7	

Table 4 Genotype of three SNPs within the EPAS1 gene and differences in SpO2 value in different populations.

	SpO2 (%) SNP sites	Wild type	Heterozygous mutation	Homozygous mutation	
Plain group	rs13419896	97.30 ± 1.36	97.08 ± 1.60	99.11 ± 1.64	
rs1868092	97.33 ± 1.37	97.00 ± 2.00	95.69 ± 3.56	
rs4953354	97.02 ± 1.53	97.92 ± 1.31	98.55 ± 1.41	
Plateau (3 days)	rs13419896	87.44 ± 3.37	87.81 ± 3.29	87.62 ± 1.60	
rs1868092	87.85 ± 3.16	86.64 ± 2.84	–	
rs4953354	87.76 ± 3.05	87.15 ± 3.41	–	
Plateau (30 days)	rs13419896	90.40 ± 2.46	90.18 ± 2.96	91.81 ± 1.41	
rs1868092	90.06 ± 2.62	91.86 ± 1.77	–	
rs4953354	90.31 ± 2.84	90.56 ± 1.24	–	
Plateau (>10 years)	rs13419896	88.75 ± 7.09	87.20 ± 5.45	92.30 ± 5.24	
rs1868092	89.62 ± 5.37	80.04 ± 3.25	86.39 ± 4.12	
rs4953354	83.00 ± 4.24	93.00 ± 2.55	84.09 ± 3.46	
Native Tibetans	rs13419896	95.00 ± 1.23	94.15 ± 2.64	94.65 ± 2.30	
rs1868092	94.33 ± 2.08	94.00 ± 2.04	94.97 ± 2.54	
rs4953354	96.00 ± 1.73	94.07 ± 2.46	94.60 ± 2.34	

Effect of EPAS1 gene polymorphism on mRNA expression

The concentration of Hb in the Han population that possessed the homozygotic plateau-adaptive genotype involving three SNPs of the EPAS1 gene was reduced, but the difference was not statistically significant; thus, it was unclear whether or not the Hb level is affected by gene expression. We therefore grouped the subjects into wild-type, plateau-adaptive heterozygous type, and plateau-adaptive homozygous type and measured the EPAS1 gene expression levels in plains male subjects before and after acute entry into the plateau. The results showed that the EPAS1 gene expression level of individuals who possessed the “A-A” homozygotic genotype at the rs13419896 locus and were living in the plains was lower than that of individuals with the wild-type and heterozygotic genotype, whereas the EPAS1 gene expression in the “A-A” plateau-adaptive homozygous genotype group was higher than that in the wild-type and plateau-adaptive heterozygous population after entering the plateau (Fig. 3).

Figure 3 Expression level of different genotypes in EPSA1 gene at Han plateau after Entering Plateau.

Stratified analysis based on different genotypes was used to analyze the expression levels of EPAS1 gene.

Differences in Raman spectra between the Tibetan and Han groups

Our previous study has demonstrated significant differences in the P50 values of Hb and oxygen affinity in different subgroups (Li et al., 2018), suggesting that, in addition to the changes in Hb concentration observed in the above studies, Hb may undergone conformational changes that are conducive to plateau acclimation. Therefore, Raman spectroscopy was used to perform Hb conformational analysis. Compared with the plains Han group, the Han group acutely entering the plateau showed a Raman band at 1,341 cm−1; this band disappeared after 30 days of plateau acclimation, accompanied by the appearance of a Raman band at 1,300 cm−1. The intensity of the Raman peak at 1,375 cm−1 increased with duration of residence in the plateau, but it decreased after 30 days of plateau acclimation, accompanied by the appearance of the Raman band at 1,355 cm−1. The group living in the plateau for a long time only displayed a characteristic peak at 1,375 cm−1, and the peak intensity was reduced. The intensity of the Raman peak at 1,546 cm−1 increased with time after entry into the plateau, reaching a maximum at 30 days of plateau acclimation; it was weakened in the group living in the plateau for a long time. The intensity of the Raman peak at 1,585 cm−1 increased with time after entry into the plateau and decreased after 30 days of plateau acclimation. The group living in the plateau for a long time displayed a characteristic peak at 1,585 cm−1, and the intensity of this peak was even weaker. The Raman intensity peak at 1,638 cm−1 increased with duration after entry into the plateau, began to decrease at 30 days of plateau acclimation and was even weaker in the group that had lived in the plateau for a long time (Fig. 4).

Figure 4 Raman spectra of RBC in Tibetans and plain population.

Comparison of the spectra recorded for the different group (A) Han population before or after entering plateau and (B) Tibetan population using 514 nm laser excitation wavelengths, showing the major band assignments in the range of 500–2,000 cm−1.

The Raman spectrum of RBCs obtained from the Tibetan population contained two bands that differed from those found in the plains group. The Raman results for six samples (60%) showed that the Raman peak intensity of the bands at 1,375 cm−1, 1,585, and 1,638 cm−1 were basically the same in Tibetans as in the plains group, but new Raman bands appeared at 1,355 cm−1, 1,546, and 1,603 cm−1. The Raman results for four samples (40%) were similar to those of the plains group, but the intensity of the peak was enhanced (Fig. 4).

Discussion

The unique natural environment of the high-altitude plateau has a great influence on human physiology, ability to work and physical and mental health. Physiological changes, such as significant increases in red blood cell counts and Hb level can improve the oxygen supply to tissues, allowing individuals to quickly adapt to the plateau. For the first time, this study conducted research on the acclimation of the Han population living in the plateau for different periods of time. The levels of Hb and SPO2 in the different population and the relationships with the gene polymorphisms and mRNA expression of the hypoxia-induction-related EPAS1 were analyzed. We also detected differences in the structure and function of Hb in the different subgroups that can be used as a means of identifying Han populations that are able to adapt to the plateau.

The results of this study showed that the Hb level of plains Han people increased significantly three days after they entered the plateau and was maintained at an average level of 176 g/L. Thus, the increased Hb levels could supply adequate oxygen to organs in the hypoxic environment, which is an important mechanism of altitude acclimation (Chen et al., 2014). In the plateau hypoxic environment, SPO2 values were significantly reduced in plains Han group; this suggests that the change in Hb content and conformation may lead to differences in physiological indicators among different populations in the high altitude environment.

The change in Hb content is closely related to the change in the genetic information for HIF. Under hypoxic conditions, EPAS1 activates the transcription of its downstream target gene EPO, thus increasing the number of red blood cell to meet the body’s oxygen supply and allowing adaptation to the plateau (Peng et al., 2017). Studies have shown that EPAS1 gene expression is also associated with EPO levels (Lee & Percy, 2011; Percy et al., 2008). In a hypoxic environment, HIF-2α, encoded by EPAS1, may accumulate in cytoplasm and then translocate into the nucleus and bind to the hypoxia response element to activate the expression of downstream genes, sunch as EPO (Petousi & Robbins, 2014) and VEGF (Sergi, 2019). Beall et al. (2010) found that 31 SNPs in the EPAS1 gene of the Tibetan population showed linkage disequilibrium and were related to the Hb concentration and that the average Hb concentration of homozygous carriers of the allele mutation was 8 g/L lower than that in heterozygous carriers. Peng et al. (2011) found that the Hb concentrations of Tibetans carrying the plateau-adaptive EPAS1 rs149594770 and rs73926265 locus genotypes were decreased by 6.15 g/L and 12.65 g/L, respectively.

This study also analyzed the causes of the changes in Hb and SPO2 levels in different subgroups from the perspective of EPAS1 gene polymorphism, and mRNA expression. First, the EPAS1 gene polymorphism study showed that the frequency of the rs13419896 A allele in the Tibetan population was 87.0%, significantly higher than its frequency in the plains population (31.5%) (P < 0.001); the frequency of the rs1868092 A allele in the Tibetan population was 76.5%, significantly higher than that of in the plains population (11.0%) (P < 0.001), and the frequency of the rs4953354 G allele was significantly higher in the Tibetan population than in the plains population (81.0% vs 13.5%) (P < 0.001). Therefore, we believe that these three SNPs have important roles in plateau adaptation and plateau acclimation. Our analysis also showed that individuals in Tibetan and Han populations with the plateau-adaptive genotypes of three SNPs had lower Hb levels than individuals of the other two genotypes.

Second, the EPAS1 expression levels in the plains Han population were found to increase after this population acutely entered the plateau and to then decrease after acclimation. In addition, the EPAS1 expression level in the Han group who had lived in the plateau for more than 10 years was significantly higher than that in the plains group. Stimulation of the expression of hypoxia-inducible factor EPAS1 by the plateau hypoxic environment is a manifestation of plateau acclimation.

To clarify whether genetic mutations in the plains population affect expression of the EPAS1 gene, we also performed a comparison of genotypes and Hb levels in different groups. The results showed that in the plains group, EPAS1 expression was significantly lower in individuals carrying the “A-A” homozygous plateau-adaptive genotype of the rs13419896 locus than individuals carrying the “G-G/A-G” genotypes. After the subjects had remained in the plateau for 3–7 days, gene expression was higher in homozygous mutants than the “G-G” wild-type and “A-G” heterozygous populations. However, there was no significant difference between different rs1868092 and rs4953354 genotypes before and after entering the plateau. This finding indicates that the Han population carrying the AA genotype of the rs13419896 locus expresses more Hb after entering the plateau. However, our test results showed that Hb levels in people with AA genotypes were lower than Hb levels in people with “G-G” or “A-G” genotypes and were not consistent with gene expression levels. We speculate that the expression of Hb under hypoxic conditions was affected by factors other than the EPAS1 polymorphism and that the single-site mutation in the gene was also not sufficient to affect the expression level of the gene.

Our previous study of Hb and oxygen affinity (P50) revealed significant differences in different among populations. In this study, Raman spectroscopy was used to investigate the Hb conformation. The region from 1,340–1,390 cm−1 is an area that is sensitive to the density of electrons in the heme ring; it is also a marker band for the redox state of heme (Kitagawa, Ozaki & Kyogoku, 1978). The increased density at the initial stage of entering the plateau may be the result of excessive release of oxygen caused by an increase in P50 and subsequent oxidation of heme. The spectrum in the 1,350–1,380 cm−1 region reflects changes in the quaternary structure of Hb (Rousseau et al., 1980). Changes in this region suggest that the stability of Hb is destroyed. The P50 value increased in the early stage of acute plateau adaptation (the 1,375 cm−1 band was enhanced); after 30 days of acclimation, the P50 value was significantly lower than the normal value of the plains population (the 1,355 cm−1 band appeared) (Li et al., 2018). In individuals who had lived in the plateau for a long time, only the 1,375 cm−1 band showed decreased intensity, this may may initiate the regulation mechanism post-compensation and regulate the P50 changes. The region of 1,240–1,300 cm−1 is related to the secondary structure of Hb (Wood et al., 2007). After 30 days of plateau acclimation, bands appeared at 1,300 cm−1, possible as a result of changes in Hb secondary structure. The hemoglobin pockets in the globin chains consist of connected α-helices, and the changes may indicate changes in hemoglobin tertiary structure and spatial conformation after altitude adaptation. The bands at 1,585 and 1,638 cm−1 are related to oxyhemoglobin (Lu et al., 2014). The Raman band changes at 3–7 days after entering the plateau. The corresponding detection result shows that P50 is increased and the organism is likely to release oxygen. However, after 30 days of high altitude acclimation and after long years of living in the plateau, the value is reduced. It is possible that the body may adjust the structure of the hemoglobin to adapt to the high altitude environment. At this time, the P50 result is reduced so that oxygen can be used for local tissue with extreme hypoxia (Wood, Tait & McNaughton, 2001). High spin state characteristic peaks of the heme near 1,546 and 1,603 cm−1, suggesting that the ions of Hb tend to show a high spin state after 30 days of altitude adaptation. In addition, the intensities of each characteristic Hb peak are weakened in Han people living in the plateau for long periods of time; this may indicate that the internal environment of the red blood cells of these individuals has undergone specific changes that result in the weakening of the characteristic peaks of hemoglobin. However, its specific mechanism of this effect will require studied.

Approximately 60% of the Tibetan population is similar to the Han people who have lived in the plateau for a long time. The intensity of the Raman peak at 1,375 cm−1 is lower in Tibetan than in the plains group, and a Raman peak appears at 1,355 cm−1. The appearance of a Raman peak in this region indicates that the changes in the structure of Hb are associated with changes in oxygen affinity. The bands at 1,585 and 1,638 cm−1 are related to oxyhemoglobin. The reduction in the intensities of the peaks in this region in the Tibetan population may be due to the adjustment of the hemoglobin structure to adapt to the high altitude environment and ultimately provide better adapt to the hypoxic environment. The corresponding reduction in P50 ensures that oxygen can be used for the hypoxic tissue. The 1,546 and 1,603 cm−1 peaks indicates that iron in Tibetan hemoglobin tends to show a high spin state, resulting in a decrease in P50. The Tibetan population has lived in a plateau environment for generations and their hemoglobin structure must be better adapted to hypoxia, but its specific adaptation mechanism should still be further explored.

Conclusion

In summary, under plateau hypoxic conditions, the Han population of the plains may rapidly acclimate to hypoxia through the EPAS1 gene polymorphism and increase EPAS1 mRNA expression and even changes in the hemoglobin conformation.

The authors thank Ying Wang for helpful discussion and review of this manuscript.

Additional Information and Declarations

Competing Interests

Author Contributions

Human Ethics

Data Availability

All authors are employed by the People’s Liberation Army (PLA).

Cuiying Li conceived and designed the experiments, performed the experiments, authored or reviewed drafts of the paper, approved the final draft.

Xiaowei Li and Jun Xiao conceived and designed the experiments, performed the experiments, analyzed the data, contributed reagents/materials/analysis tools, prepared figures and/or tables, authored or reviewed drafts of the paper, approved the final draft.

Juan Liu performed the experiments, authored or reviewed drafts of the paper, approved the final draft.

Xiu Fan performed the experiments, contributed reagents/materials/analysis tools, prepared figures and/or tables, approved the final draft.

Fengyan Fan performed the experiments, contributed reagents/materials/analysis tools, authored or reviewed drafts of the paper, approved the final draft.

Huifen Lei performed the experiments, prepared figures and/or tables, approved the final draft.

The following information was supplied relating to ethical approvals (i.e., approving body and any reference numbers):

The Institutional Review Board of Air Force Medical Center, PLA approved this research (2017-05-YJ01).

The following information was supplied regarding data availability:

Data is available at Zenodo: Cuiying Li. (2019). Genetic changes in plateau [Data set]. Zenodo. http://doi.org/10.5281/zenodo.3333456.

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
