# Peer review of "Genetic changes in the EPAS1 gene between Tibetan and Han ethnic groups and adaptation to the plateau hypoxic environment"

_PeerJ, doi:10.7717/peerj.7943_

## Round 0.1 · original submission · Major Revisions

Your manuscript describes an interesting topic of local adaptation leading to differences between ethnic groups residing at different altitudes. Its finding can have important implications for medicine. However, both reviewers found several issues that need to be corrected before the manuscript can be accepted. The reviewer 1 has annotated your manuscript, and the reviewer 2 has provided a list of relevant literature that can be used for interpretation of your results. I hope that you will be able to revise and resubmit your manuscript withing the next 55 days.

Reviewer 1 ·

Basic reporting

It fails

Experimental design

It fails

Validity of the findings

Some novel findings but would need to be rewritten, see my detailed critiques.

Additional comments

see highlighted portions of the manuscript with specific comments and suggestions fair revision

Annotated reviews are not available for download in order to protect the identity of reviewers who chose to remain anonymous.

·

Basic reporting

The group of the Air Force Medical Center, PLA, Beijing, China, studied genetic changes in the EPAS1 gene between Tibetan and Han ethnic groups and adaptation to the plateau hypoxic environment. Genomics, transcriptomic microarray, and whole-genome linkage analysis have revealed that the hypoxia-inducible factor (HIF) plays an important role in altitude hypoxia adaptation. Studies have shown that HIF is widely expressed in mammals and that it plays a key role in hypoxia and can promote high-altitude adaptation. Recently, it has been emphasized on endothelial PAS domain protein 1 (EPAS1) (Sergi C, 2019; https://www.ncbi.nlm.nih.gov/pubmed/31015364). In fact, a single nucleotide polymorphism (SNP) in EPAS1 is closely related to the low level of hemoglobin (Hb) in Tibetan populations. EPAS1 expression plays an important role in oxygen metabolism and sensing. The authors found that under plateau hypoxic conditions, the Han population of the plains may rapidly acclimate to hypoxia through the EPAS1 gene polymorphism and increase EPAS1 mRNA expression and even changes in the hemoglobin conformation. The manuscript is well structured, but typos and grammar are deficient. The references have different citation styles. The association of EPAS1 with congenital heart disease may be considered in the discussion.

Experimental design

The experimental design is appropriate and research questions are well defined. The method of selection of patients for Raman spectroscopy is not clear and needs to be clarified (e.g., random number generator).

Validity of the findings

All data seem to be robust and statistically sound. Conclusions are adequate, although further studies on experimental studies may be important for the future. In particular, the discussion evidenced on dysmorphology of EPAS1 and the role of epigenetics may be emphasized in the manuscript. Please see https://www.ncbi.nlm.nih.gov/pubmed/31015364.
The adaptation of the Han population to the plateau may have important consequences for future generations.

Additional comments

To the best of my knowledge, I found the manuscript an interesting one that will be important to medicine. I hope that the authors may consider my suggestions as indicated above to improve the manuscript.

---

## Round 0.2 · accepted · Accept

Your manuscript presents an interesting problem of finding genetic mechanism of altitude adaptation. In future, it will be very interesting to extend your analysis to other population groups, such as highland and lowland populations of Caucasus.

·

Basic reporting

Improved manuscript!

Experimental design

Good design and good methodology.

Validity of the findings

Valid data that may be very useful for studies on VEGF interaction.

Additional comments

Thank you for improving the manuscript.